# Valve cells are crucial for efficient cardiac performance in *Drosophila*

Christian Meyer[1], Achim Paululat[1,2]*

**1** Department of Biology/Chemistry, Zoology & Developmental Biology, Osnabrück University, Osnabrück, Germany, **2** Center of Cellular Nanoanalytics (CellNanOs), Osnabrück University, Osnabrück, Germany

* achim.paululat@uos.de

## Abstract

Blood flow in metazoans is regulated by the activity of the heart. The open circulatory system of insects consists of relatively few structural elements that determine cardiac performance via their coordinated interplay. One of these elements is the intracardiac valve between the aorta and the ventricle. In *Drosophila*, it is built by only two cells, whose unique histology represents an evolutionary novelty. While the development and differentiation of these highly specialised cells have been elucidated previously, their physiological impact on heart performance is still unsolved. The present study investigated the physiological consequences of cardiac valve malformation in *Drosophila*. We show that cardiac performance is reduced if valves are malformed or damaged. Less blood is transported through the heart proper, resulting in a decreased overall transport capacity. A reduced luminal opening was identified as a main reason for the decreased heart performance in the absence of functional valves. Intracardiac hemolymph flow was visualised at the valve region by microparticle injection and revealed characteristic similarities to valve blood flow in vertebrates. Based on our data, we propose a model on how the *Drosophila* intracardiac valves support proper hemolymph flow and distribution, thereby optimising general heart performance.

**Data availability statement:** All relevant data are within the paper and its Supporting Information files.

## Author summary

Blood flow in metazoans is regulated by the heart's activity. Insects possess an open circulatory system with a few structural components that collectively influence cardiac performance. A key element is the intracardiac valve between the aorta and the ventricle, which in *Drosophila* consists of only two specialised cells, representing an evolutionary innovation. While the development of these cells has been studied, their physiological impact on heart function is less understood. This study investigates the consequences of cardiac valve malformation in *Drosophila*. Our findings indicate cardiac performance declines when the valves are malformed or damaged, leading to decreased blood transport and lower overall capacity. A reduced luminal opening contributes significantly to diminished heart performance without functional valves. Moreover, the visualisation of intracardiac fluid flow using microparticle injection reveals similarities to blood flow in vertebrates. Based on these insights, we propose a model detailing how intracardiac valves optimise fluid flow and enhance insect heart performance.

**Funding:** This work was supported by grants from the Deutsche Forschungsgemeinschaft to A.P. (PA 517/13-1 & PA 517/13-2, SFB 1557). We also acknowledge the support of the Open Access Publishing Fund of the Osnabrück University. The funders had no role in study design, data collection and analysis, decision to publish, or preparation of the manuscript.

**Competing interests:** The authors have declared that no competing interests exist.

## Introduction

Cardiac valves control blood flow directionality, thereby enhancing pumping against higher pressures and gravity, thus boosting cardiac performance [1–7]. The morphology and function of cardiac valves differ significantly among animals due to adaption to environmental conditions. For example, mammalian multicellular valves, separating the heart chambers, consist of connective tissue with collagen and elastin as main molecular constituents. In contrast, venous valves constitute cellular flaps lined with a thin matrix [8].

In vertebrates, and presumably in all animals harbouring a closed circulatory system, the directionality of blood flow is regulated by intracardiac valves, which are constructed as flap-like structures that open and close the luminal space of the vessel. However, certain disease symptoms are grounded in the malformation of cardiac valves. For example, patients suffering from Marfan syndrome exhibit myocardial infarction caused by the failure of proper valve function, resulting in the narrowing of the heart lumen (stenosis) and consequently compromising heart function [9]. Furthermore, aortic atherosclerosis leads to rigidification of the connective tissue and formation of calcium deposits on the valves in all mammals, birds, reptiles and fish and may finally also lead to stenosis [10–14]. As a consequence, severe cardiac valve malformations dramatically impact cardiac output and overall health in general [15].

Invertebrate hearts, like in Diptera, exhibit rather simple valves. They are formed by simple structures like muscle pillars, large spongy cells or cellular pads [16–18]. *Drosophila*, like all insects, possess an open circulatory system. The movement of the hemolymph, the insect blood, is driven both by the activity of cardiac muscles and by contraction of the body wall muscles. Additionally, heartbeat reversal, changing the flow direction of hemolymph under certain physiological conditions, has also been described in many insects [19–23]. As a consequence, the intracardiac valves situated at the transition of the wide-luminal ventricle and the narrow-luminal aorta, must function as two-way gates, which open and close for streaming in both directions. In the present study, we investigated the role of *Drosophila* intracardiac valves in hemolymph distribution, heart performance and the extent to which they contribute to the control of flow properties within the heart. Only in recent years, light has been shed on the complex biogenesis and function of these intracardiac valves and their unique ultrastructure [24–28]. Our previous work introduced various tools for studying the *Drosophila* valve cells. We identified two valve-specific reporter lines [27,28] and several critical genes, which, upon down-regulation or expression as constitutively active variants, induce different types of malformations in valve cells, without affecting neighbouring cardiac cells [27]. We now improved and extended previously established methods to analyse hemolymph flow in living and intact animals. The combination of different imaging methods allowed us to analyse the physiological consequences of valve cardiac malformation in *Drosophila*.

Based on our experimental data, we postulate a model in which the intracardiac valves evolved in *Drosophila* to ensure luminal closure upon heart beating accompanied by a high deformation capability needed to fully open the heart lumen. These features allow efficient hemolymph distribution in the animal. A reduction of the luminal opening, caused by valve damage in mutants, reduces the amount of circulating hemolymph, reduces overall fitness and negatively impact hemolymph dynamics in the fly heart.

We, therefore present our current study as an essential first step for future work with a focus on analysing the physiological significance of the insect heart for the animal as a whole.

## Results

Previously we demonstrated that valve cells utilise existing endosomal pathways to generate large intracellular membranous cavities, the valvosomes (Fig 1C and 1D; [27]). Genetic

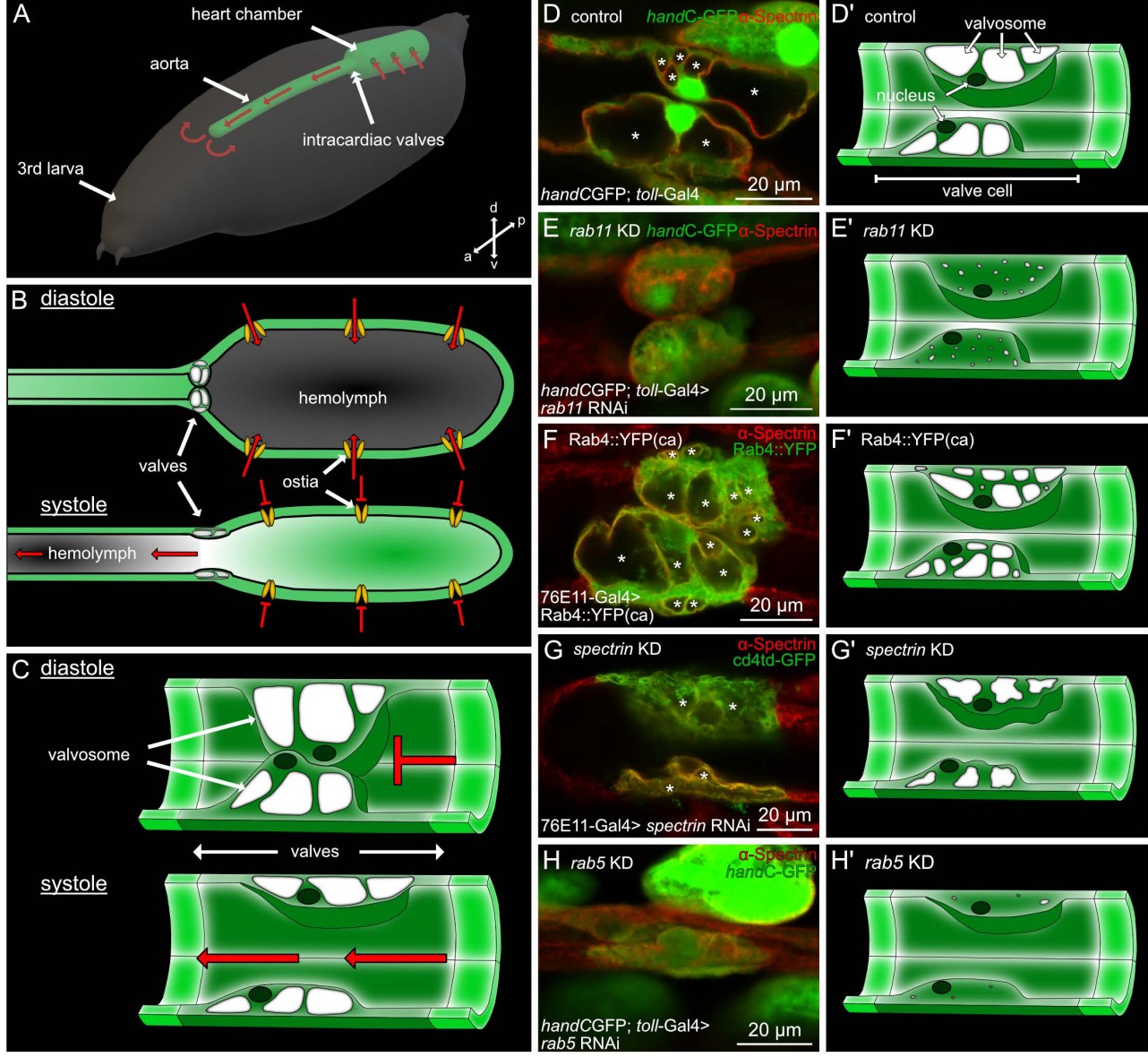

**Fig 1. Malformations of cardiac valves in *Drosophila*.** (A) Open circulatory system in the *Drosophila* larva. Hemolymph enters the heart chamber via pairs of ostia cells, and is pumped anteriorly through the aorta. It then circulates in the open body cavity and reenters the heart again. The intersection of aorta and heart chamber is separated by a pair of intracardiac valves. The direction of the hemolymph flow is depicted by red arrows. (B) During diastole hemolymph (depicted in grey) enters the heart chamber via ostia cells. Valve cells constitute a roundish shape and seal the heart lumen. Upon systole, the heart chamber contracts, valve cells possess an elongated shape opening the heart lumen and hemolymph is pumped anteriorly (see S1 Video). (C) Schematic illustration of cardiac valves during diastole and systole. Upon diastole, valve cells press their cell bodies into each other and seal the heart lumen. Upon systole valves become elongated, thereby opening the heart lumen. (D, D') In control animals, valves are characterised by a large roundish cell body and 2-4 large valvosomes. (E, E') Upon knockdown of *rab11* valvosomes are almost absent from valves. (F, F') Ectopic expression of a constitutively active form of Rab4 leads to an increased number of smaller valvosomes. (G, G') Knockdown of alpha-*spectrin* leads to a collapse of valve cells and valvosomes. (H, H') Inhibition of *rab5*-mediated endocytosis results in significantly smaller valves cells and the absence of valvosomes.

manipulation of the Rab5-dependent early endocytosis, the Rab4/Rab11-dependent endo-somal recycling pathways or components of the cytoskeleton, like Spectrin, affect proper valvosome biogenesis and lead to different valve phenotypes (Fig 1; [27]). Phenotypes are characterized by an overall reduction of valve cell size, a reduction in the number of

characteristic valvosomes per valve cell, as well as an overall reduction of the size of the valvosomes. For instance, knockdown of *rab11* leads to a reduction of valve size by 50% and the total number of valvosomes per cell is reduced from 3-4 valvosomes to 1 or the complete absence of this structure occurs. These morphological changes come along with changes in the valves operating mode, as damaged valves show reduced deformation capabilities upon heart beating [27].

Together with our valve-specific Gal4 driver lines, this now gives us the unique opportunity to manipulate valve cells without affecting other cardiac cells. Thus, for the first time, we can experimentally analyse the influence of valve cells on hemolymph circulation and heart performance in insects.

## Damaged valves lead to a reduction in cardiac performance

To investigate cardiac output and hemolymph pumping efficiency, we performed dye-angiography. Dye angiography has been previously established to visualise and measure hemolymph streaming in *Drosophila* and *Anopheles* (Fig 2A) [29–33]. The method relies on the injection of a tracer dye into the abdominal body cavity of living specimens. The dye is sucked into the heart chamber and pumped anteriorly upon rhythmic heart beating. We used the accumulation of the dye in the head region of the animal as read-out for cardiac efficiency (Fig 2) [30,31].

As previously reported, changing the expression levels of Rab4, Rab5, Rab11 or alpha-Spectrin, specifically in valve cells, led to malformed valves to different degrees. We measured the cardiac performance in control mutant animals, and found significantly less dye accumulating in the head region in any of the valve mutants compared to the control (Fig 2D-J). Accumulation of the tracer was reduced by 40% to 50% in *rab11* knockdown, *alpha spectrin* knockdown and Rab4 overexpressing animals (Fig 2J). Concurrent to previous findings, knock-down of *rab5*, which leads to the most severe damage in valve cell morphology (Fig 1H), also caused the most dramatic decline in pumping efficiency by 80% to 90% (Fig 2J and 2I). Therefore, our results clearly demonstrate that reduced cardiac output and hemolymph pumping efficiency is due to damaged cardiac valves. Why is this the case?

## Luminal closure and hemolymph velocities

After analysing the impact of valve anatomy on the global motion of hemolymph in the body, we next addressed the motion of hemolymph within the heart tube itself. The time-restricted luminal closure of the heart lumen, ensuring the controlled hemolymph filling of the heart chamber, constitutes the most fundamental function of active cardiac valves in the *Drosophila* heart [26–28] (Fig 1A-C). In *Drosophila,* hemolymph is transported by the rhythmic transport of hemolymph from the heart chamber into the anterior aorta and afterwards into the open body cavity [30]. As a result of the rhythmic open/close cycles of the valve, hemolymph is not transported in a continuous current, but rather in individual units (Fig 3A-C). In order to estimate the flow pattern within the heart, we precisely selected different regions of interest (ROIs) within the aorta, and monitored the passing of single tracer units over time. This allowed us to estimate luminal closing capabilities of the valve, the velocity of the transported hemolymph, as well as the hemolymph "package" size [30].

Measuring image intensities in single ROIs resulted in periodic oscillograms, reflecting the motion of the tracer and the periodic open/close cycles of the valve. In control animals, intensity values periodically alternate between peaks of high and low intensity, reflecting the ability of valves to close the luminal space between heartbeats. Although the bulk motion of hemolymph was dramatically affected, we found that hemolymph transport within the heart

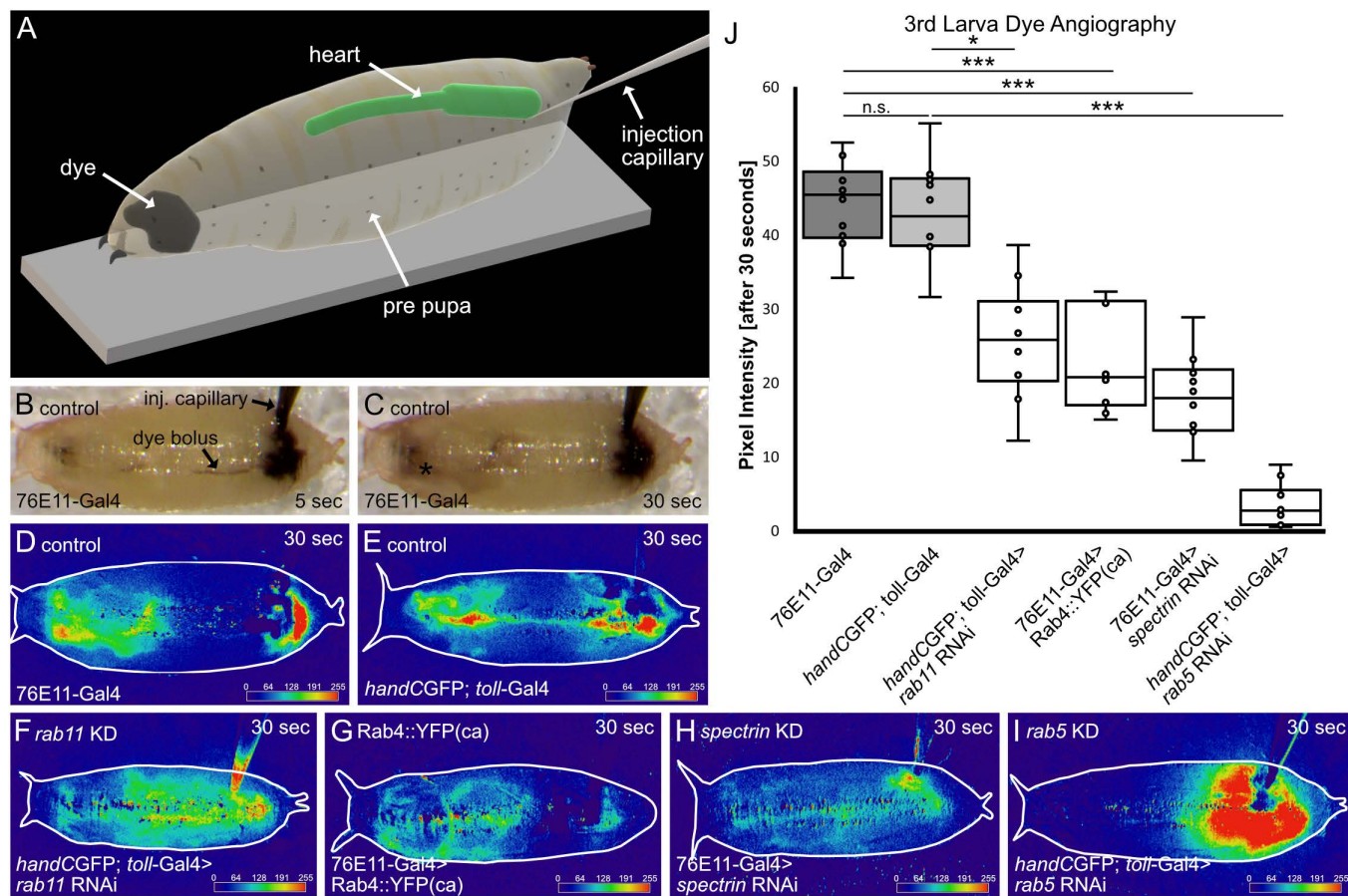

**Fig 2. Cardiac performance and hemolymph distribution are significantly reduced upon valve malformation.** (A) Scheme illustrating the methodology of dye angiography analysis. Black ink is injected in the posterior end of early white prepupae. Tracer dye accumulates over time in the anterior region due to the continuous pumping activity of the heart (see S2 Video). (B, C) Light microscopic illustration of a control animal 5 sec and 30 sec after injection, respectively. Injected dye accumulates in the anterior region (asterisk). (D-I) Digital subtraction and colour-coded pixel intensity of the tracer dye in controls (D, E) and valve mutant animals (F-I). (J) Statistical analysis of the dye angiography. After 30 seconds significantly less dye was transported to the anterior end in valve mutants, compared to control lines. *N*= 10 animals per genotype. Two-tailed Student's *t*-test *$P < 0.05$, ***$P < 0.001$.

appeared normal in most valve mutant lines, showing nearly identical efficiency (Fig 3D-H). Nevertheless, based on pixel intensities the amount of tracer dye transported by each contraction cycle was reduced in these valve mutants, explaining the impact on the bulk motion measured before (Fig 3J). Oscillograms from *rab11* knockdown animals showed similar peaks of high and low intensities, indicating proper time-restricted luminal closure of the valves but in addition passages of low intensities are present, representing cardiac arrests (Fig 3F). Most dramatically, oscillograms obtained from *rab5* mutant animals, which display the most severe malformed valves, showed no periodic change of intensity at all (Fig 3J). Thus, these animals completely fail to sustain an efficient hemolymph transport within the heart. Over time, the aorta was only very slowly filled by the dye, most likely driven by diffusion (Fig 3I). As a result, we conclude that less hemolymph is transported per heartbeat if valves are malformed, leading to a reduced hemolymph distribution (Fig 2J).

## Streaming velocity inside the heart

In humans it has been shown that cardiac and vascular valves control aortic blood velocity. In healthy people, aortic blood velocities vary under different circumstances. For instance,

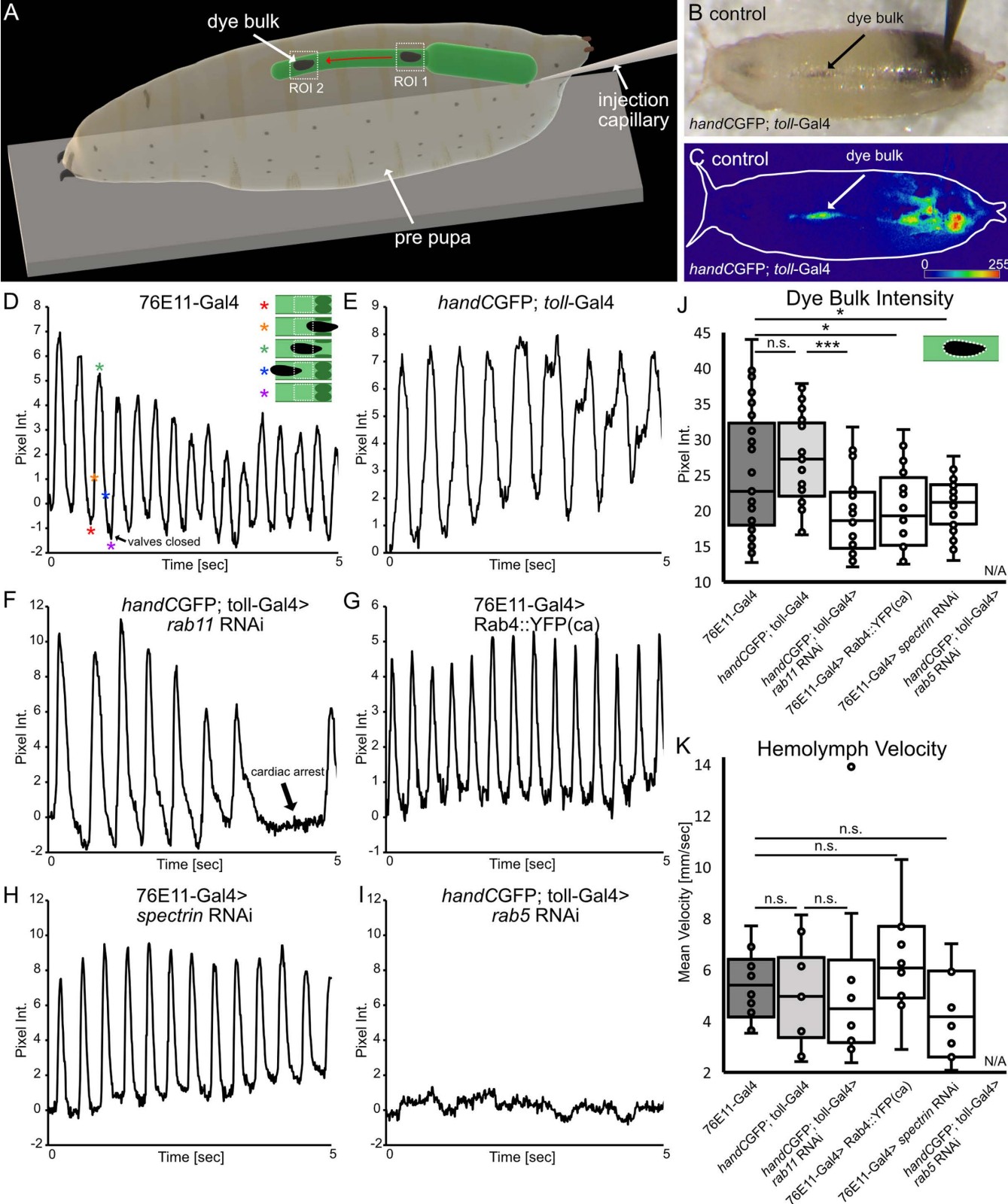

**Fig 3. Valve closing capability and hemolymph flow analysis.** (A) Scheme illustrating experimental setup for valve closing efficiency and hemolymph flow analysis. A region of interest (ROI 1) is set anteriorly to cardiac valves. Pixel intensity in ROI 1 increases if hemolymph packages pass and decreases, if valves close after each heartbeat, respectively. Hemolymph velocity was calculated using two regions of interest (ROI1 & ROI 2) in the aorta. The respective time

for one hemolymph tracer packages reaching peak pixel intensity in the two ROIs was calculated to the distance of the two ROIs. (B) Brightfield illumination showing injected white pre-pupa and hemolymph flow in the aorta. (C) Digital subtraction and color-coded pixel intensity, same animal as in (B). (D-I) Valve closing capability. Upon heartbeat hemolymph packages pass ROI1 (located anteriorly to cardiac valves), reaches peak intensity and falls down to a baseline again, indicating hemolymph transport and proper valve luminal closure in control animals and in *rab11* knockdown, *spectrin* knockdown and Rab4 overexpression lines (D-H, Scheme in D highlights characteristic timepoints of the pumped dye package passing the ROI, asterisks represent respective points in the oscillogram. (I) In *rab5* knockdown animals no active hemolymph transport was detected. (J) Analysis of hemolymph flow in the aorta. Mean pixel intensity of the dye package is significantly reduced upon valve malformation, indicating less hemolymph is pumped (Scheme illustrates dye package in the aorta and its area analyzed). For *rab5* knockdown animals no tracer package was formed and could be analyzed. $N= 5$ animals per genotype. (K) Hemolymph aorta velocity is unaffected if valves are malformed. For *rab5* knockdown animals, hemolymph velocity could not be determined. $N= 10$ animals per genotype. (J) Two-tailed Student's *t*-test, (K) *rab11* KD *Mann-Whitney* test $*P < 0.05$, $**P < 0.01$, $***P < 0.001$.

at rest peak velocity is about 1 m/sec, which can increases to more than 3 m/sec during exercise [34]. In addition, aortic velocity measurements have been used as readout for the classification of the severity of stenosis phenotypes [35]. This led us to assume that malformations in the *Drosophila* valve cells may also lead to aberrant aortic hemolymph velocities. Thus, we measured aortic hemolymph velocities based on the oscillograms obtained from the dye angiography experiments (Fig 3A and [30]). We found that cardiac valve malformation does not impact aortic hemolymph velocities (Fig 3K). Aortic velocities in healthy animals was estimated as ~5,2 ± 1,5 mm/sec, while upon valve damage aortic velocities ranged from ~4 to 6 ± 2,5 mm/sec (Fig 3K). These results are comparable to aortic velocities previously reported [30].

## The luminal distance in the open state is significantly reduced in valve mutants

Previously we demonstrated that the deformability of the valve cells is critical for their mode of action. We speculated that interfering with the four genes regulating valvosome formation, *rab4*, *rab11*, *alpha-spectrin* or *rab5*, may lead to a reduced deformability of these cells upon heart beating [27]. A reduced deformation capability of valves is often associated with a highly reduced luminal opening upon systole and thus opening duration time [36,37]. For this reason, we investigated the closing characteristics and motility of the healthy and malformed valves in semi-intact heart preparations in more detail (Fig 4A-F). We found that the luminal distance in the peak open state, when the hemolymph is pumped anteriorly, is significantly reduced in all valve mutants analysed (Fig 4Q). Interestingly, valve cells of *rab5* knockdown animals were unable to close the heart lumen in semi-intact preparations properly. Therefore, valve opening time could not be estimated. However, measuring the time from the initial luminal opening until the end of a luminal closing, which includes the peak opening phase, we found no differences between control groups (Fig 4G-K and 4R) and valve mutants (Fig 4L-P and 4R). Our results show that the duration of the overall opening time remains unaffected, but the maximal luminal opening during peak opening phase is considerably reduced. This indicates that malformed valve cells impact the ratio of the different opening and closing phases of the valve during a complete heart cycle, but do not influence heart rhythm.

## Presence of intact valve cells is critical for controlling physical streaming parameters inside the heart tube

In order to analyse the dynamics of hemolymph streaming in living insects, several methods have been established. These include particle injection into intact *Drosophila* and *Anopheles* specimens [26,38,39], microbubble injection into *Schistocerca americana* [40] or tracking of GFP-labelled hemocytes in *Drosophila* [41]. However, continuous darkening of the cuticle

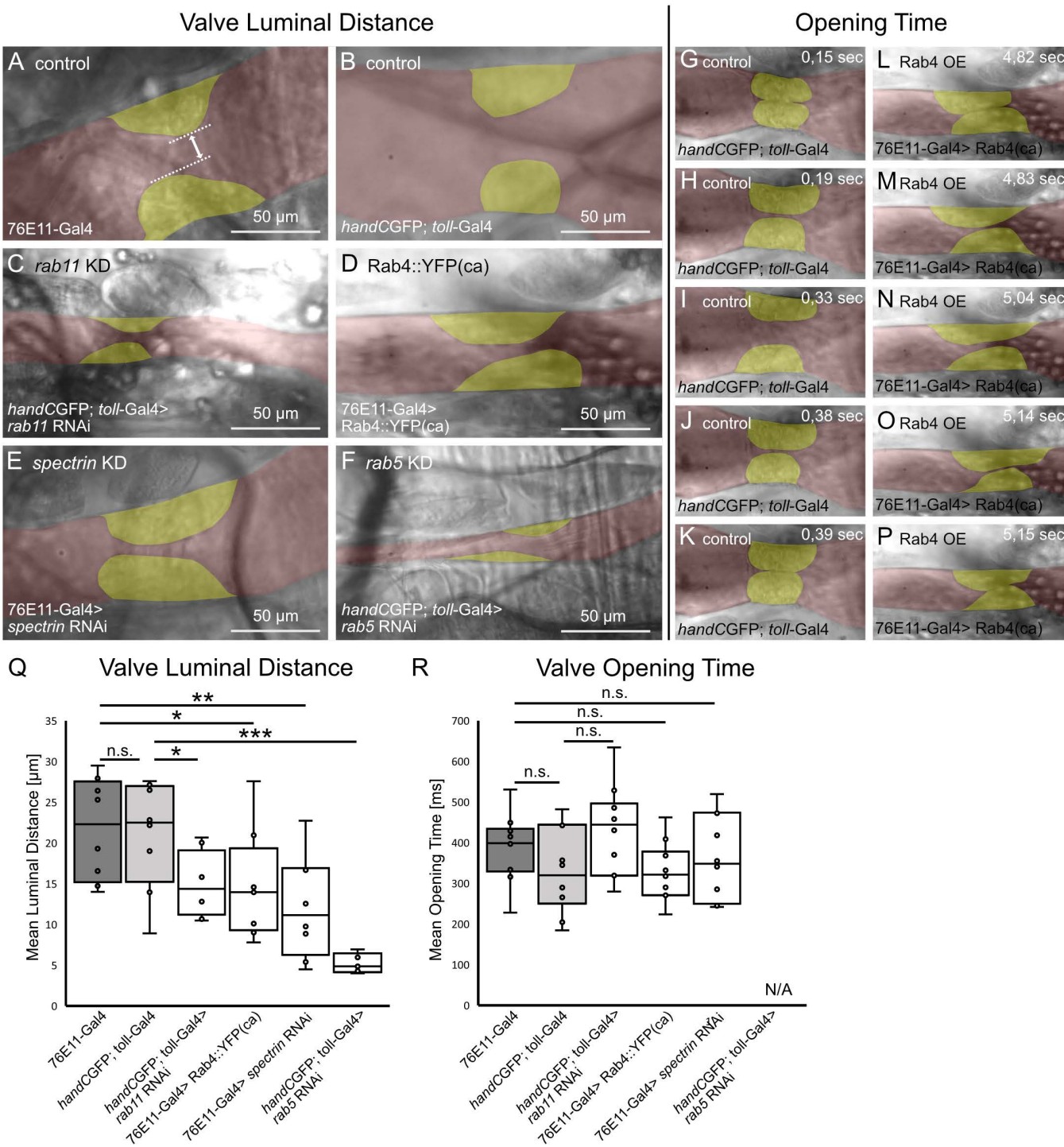

**Fig 4. Valve luminal distance and opening time upon valve malformation.** (A-F, Q) Valve luminal opening is significantly reduced, if valves are malformed, but the time of valves in the open state is similar to control animals (R). (G-K) Representative frames from highspeed videos of dissected 3rd instar larva of control and Rab4 overexpression animals (L-P), showing one opening cycle of valves. Valves depicted in yellow, heart lumen depicted in red. *N*= 10 animals per genotype. Two-tailed Student's *t*-test *P < 0.05, **P < 0.01, ***P < 0.001.

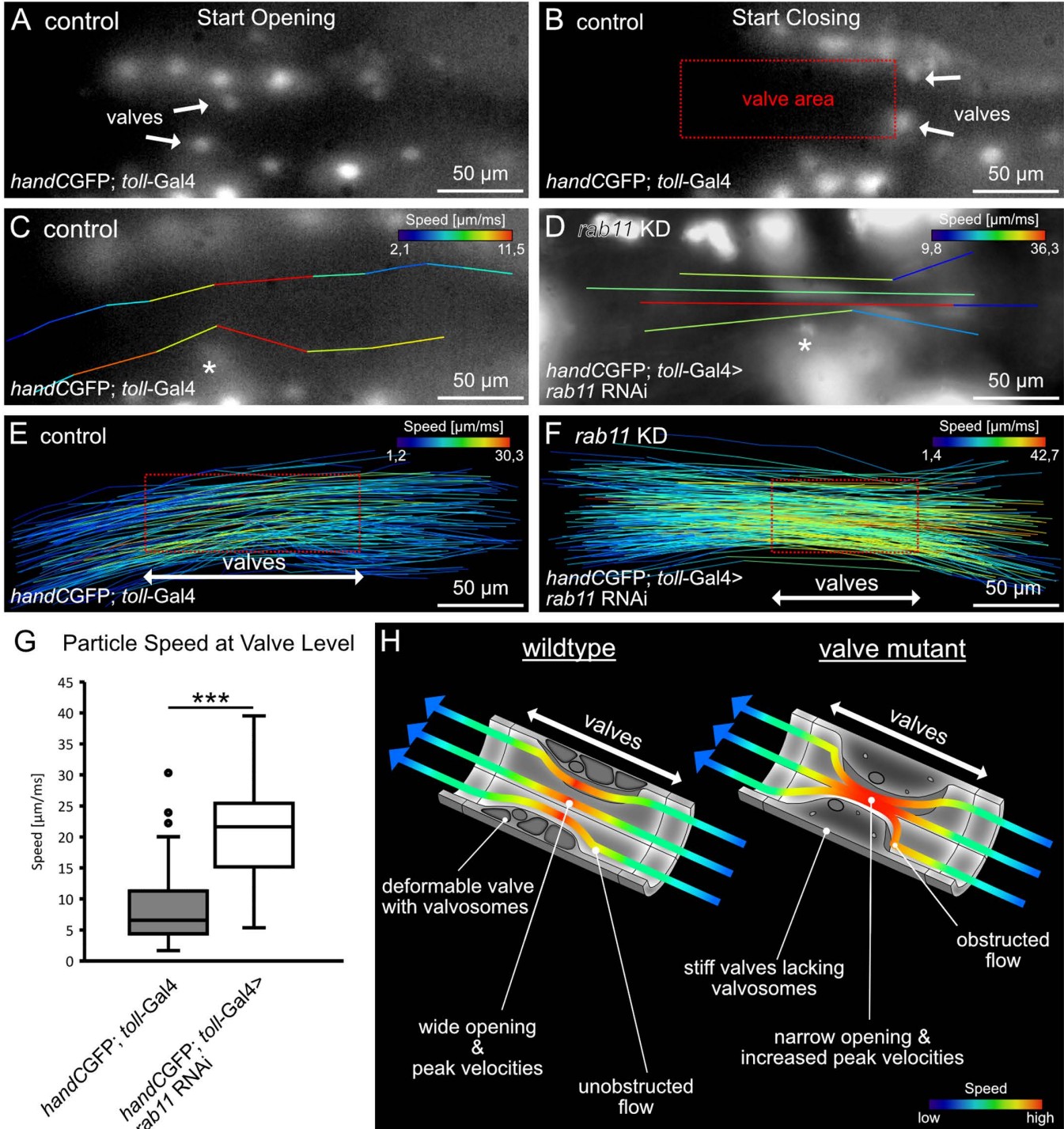

**Fig 5. Particle streaming at the valve.** Selected frames of a series highlighting valves position upon heart beating (A-B). Valves elongate and shorten upon heartbeating leading to a distinct narrowed lumen in the heart tube (Box in B). Tracks of injected particles in the heart tube passing control (C) and mutant valves (D) (Asterisk indicates position of one valve). Particles flowing in the outer lumen get partially deflected by valves physical presence, while particles in the center of the lumen are unaffected. All particles reach highest velocity when passing the valve cells (C-D). (E-F) Scheme showing speed velocities of all tracks analyzed. Red box represents the area occupied by valve cells upon heart beating. (G) Particle speed at the position of the valves is highly increased upon valve damage. *Mann-Whitney* test ***$P < 0.001$. (H) Model illustrating hemolymph flow in wildtype and valve mutants. Wildtypic valves are highly deformable and open the heart lumen, allowing unobstructed flow. In contrast, in valve mutants heart lumen is narrow and flow is obstructed. Higher streaming velocities are present upon valve damage.

and massive line-of-sight obstructions caused by adjacent tissues impair the precise imaging and a detailed analysis of the dynamic hemolymph flow at the level of valves. Therefore, we improved our hemolymph imaging methods significantly by using smaller, fluorescent particles that were injected into transgenic reporter lines expressing GFP in cardiomyocytes, including the valve cells (Fig 5). This allowed us to visualise and to analyse hemolymph streaming properties in detail.

Our live imaging approach revealed that due to valve cells elongation and shortening in anterior-posterior direction, a defined narrowed area is present during systole. (Fig 5A-B). We found that particles displayed highest speeds when passing this narrowed area between the valve cells. Moreover, we identified different types of flow-tracks with our measurements. Particles that stream in the periphery of the heart lumen became partially entangled by the physical presence of valves (Fig 5C). By contrast, particles that stream in or near the centre of the heart lumen remain unaffected (Fig 5C). However, we found that all particles analysed reach highest flow speeds when travelling through the area spanned by the intracardiac valves (Fig 5E). Similar observations were made for animals with damaged valves (Fig 5D and 5F-H). Particles got obstructed by less deforming valves and reached highest speeds when passing the naworred area spannend valves (Fig 5D-F). In fact, particle speed was significantly increased by ~30 percent when valves were damaged (Fig 5G, comparison of highest single speeds measured).

From our experimental data we propose a model where intraluminal hemolymph streaming at the valve level is highly affected by the physical presence of the valves themselves (Fig 5H). In wildtype flies hemolymph directionality will only be partially altered when passing the narrowed area spanned by deforming valves. Furthermore, hemolymph velocity will increase from posterior to anterior and reach highest speeds at the position of valves before slowing down again anteriorly. Upon damage the valve's deformation capability is reduced, resulting in a more obstructed particle flow and significant increase in particle peak velocities at the valve position compared to controls.

## Cardiac morphology is unaffected upon valve malformation

Cardiac diseases often have an impact on the entire vascular system, especially the morphology and function of the contractile hearts and the valves are affected [12,42–44]. In addition, heart morphology is one of the most important aspects in terms of the regulation of cardiac output, because cardiac output is the product of heart rate and stroke volume. Since we observed a reduced cardiac output, hemolymph distribution, and luminal opening in valve mutants, we wondered whether this is related to an effect on overall heart morphology.

First, we measured the area of the heart lumen in the cardiac ventricle in dissected and Phalloidin-stained animals in the abdominal segment five (A5) (Fig 6A-G). To ensure maximum opening of the heart, we treated the animals with MgCl2 prior to dissection, causing the complete relaxation of all somatic muscle cells. We could not find any differences between the different genotypes compared to controls (Fig 6G), except for the *rab5* knockdown which led to a reduced heart size (Fig 6F-G)). To exclude potential differences in myofiber morphology, which may affect cardiac contraction capability and thus cardiac output, we analysed the orientation and density of single myofibers of cardiomyocytes in a ROI in A5 (Fig 6A-F, 6H and 6I). With the exception of the downregulation of *spectrin*, which leads to a slight increase in myofiber density (Fig 6E' and 6H), there were no differences in myofiber orientation and density in all animals investigated (Fig 6H and 6I). These results indicate that valve malformation has no impact on overall heart morphology, and thus reduced cardiac output and hemolymph distribution in valve mutants seems not connected to an altered heart morphology.

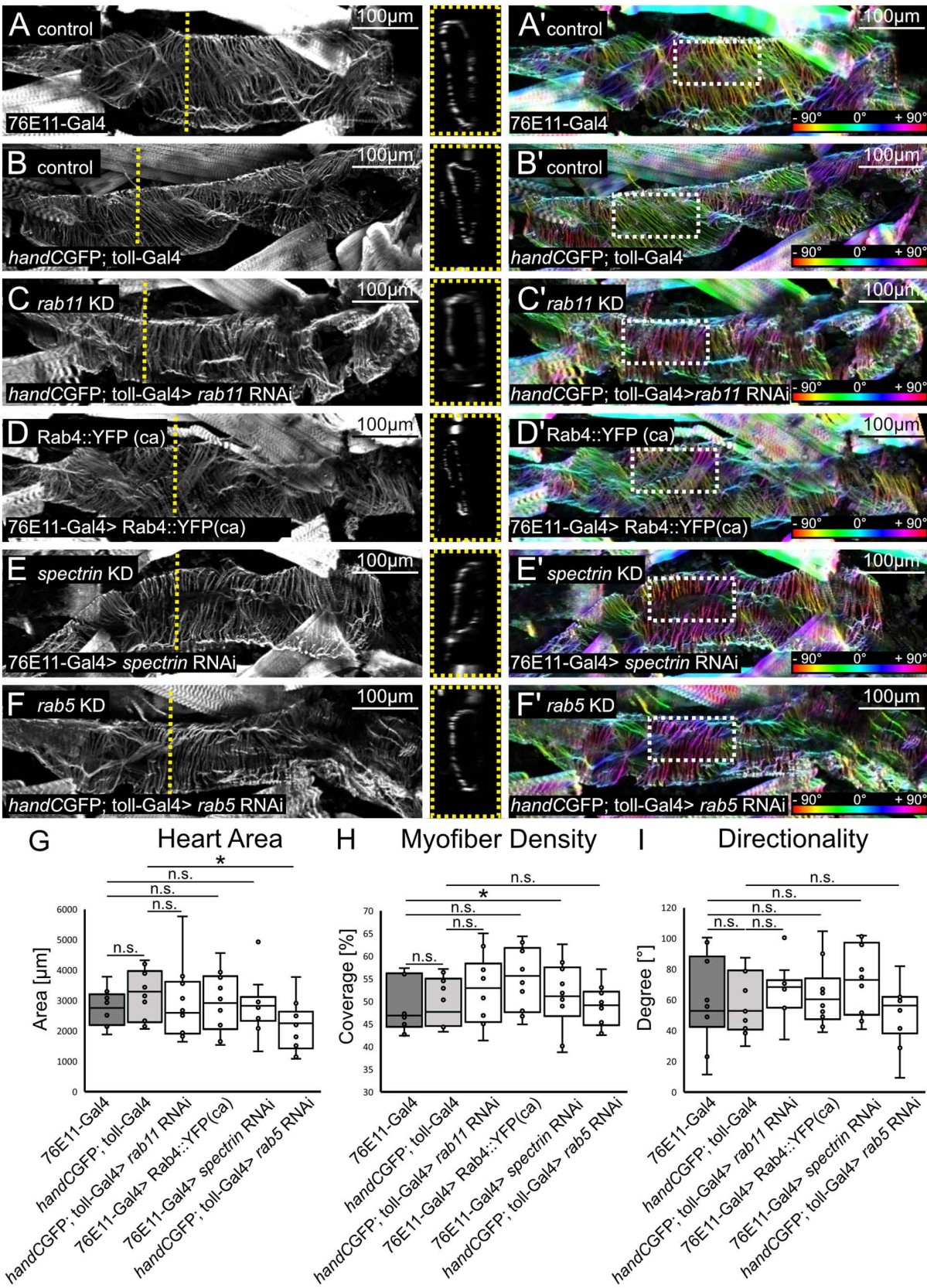

**Fig 6. The overall heart morphology is not altered if valves are malformed.** (A-G) Heart area size was unaffected after knockdown of *rab11*, *spectrin* or ectopic expression of *rab4*, but slightly reduced in *rab5* knockdown animals (F-G). Dotted yellow line indicates the point of orthogonal view of heart area measurements in abdominal segment five (boxed yellow). (A'-F') Color-coded images of respective hearts showing orientation and density of myofibers analyzed in a 100μm x 50μm ROI in abdominal segment 5 (A5, dashed white boxes). (H) Myofiber density is slightly increased upon ectopic expression of Rab4 (D') but is unaffected and similar to controls if *rab11*, *rab5* or *spectrin* was downregulated (C', E',F',). (I) Orientation and directionality of myofibers were similar between all groups investigated. *N* = 10 animals per genotype. Two-tailed Student's *t*-test *$P < 0.05$, **$P < 0.01$, ***$P < 0.001$.

## Cardiac parameter in valve mutants

In severe cases of cardiac atherosclerosis, autonomic control of resting heart rate is impaired in patients with aortic stenosis [45]. Thus, we were wondering if cardiac parameters in *Drosophila* were altered due to a different valve morphology and deformation as well. We performed heart analysis of third instar larvae aorta utilising semiautomatic optical heart analysis (SOHA) [46–48]. We found that heartrate, heartperiod, diastolic and systolic intervals of valve mutant lines were comparable to controls (Figs 7 and S1). This indicates, that the reduced cardiac performance and hemolymph output we observed in our dye injection experiments is not related to any of these parameters.

However, upon knockdown of *rab11* we found an increase in the arrhythmia index (Fig 7). Interestingly, such cardiac arrests can also be seen in our oscillograms of injected tracer dye (Fig 3F). Ectopic expression of Rab4 did not alter cardiac parameters, however, knockdown of *spectrin* slightly impacts the fractional shortening of the heart (Fig 7). Most dramatic changes on valve morphology and cardiac parameter can be found after blocking endocytosis by knockdown of *rab5* (Figs 1H and 7). Affected animals possess a prolonged heart period and diastolic and systolic intervals, as well as a decreased heartrate, arrhythmia index, fractional shortening, diastolic and systolic diameter (Fig 7) probably related to not fully functional valves not closing the heart lumen and transporting hemolymph efficiently (Figs 2J, 3K and 4R).

In sum, our results indicate that the reduction of cardiac output und hemolymph transport in animals with damaged valves cannot be explained by respective cardiac parameters analyzed. However, if the valves modes of operation and luminal closure is impaired, as in the case of *rab5* downregulation, cardiac parameters are highly impacted.

## Cardiac valve malformation affects larval endurance

Since we found a dramatic reduction in the bulk motion of hemolymph, as a consequence of severely impaired valve morphology, we asked whether this might have an effect on the fitness of the larva. Therefore, we performed a larval crawling assay, a reliable method to study *Drosophila* larva's fitness [49–51]. With the exception of the knockdown of *rab11*, all valve mutant animals exhibited a reduced crawling ability. The animals were found to crawl shorter distances during one minute of observation (Fig 8A and 8B), and often showed erratic and less directed motion away from the starting point (Fig 8C and 8E). In addition, larvae crawled at a significantly lower speed compared to controls (Fig 8D).

In addition to a directed pumping by the dorsal vessel, hemolymph can be distributed through the body by muscle contractions and volume changes of the abdomen of the insect [52,53]. To asses larval body wall contraction, we monitored the shape of individual larvae upon crawling. We used ImageJ *Shape Descriptors* measurement function, to estimate roundness of crawling larva (Fig 8F-G). This type of measurement is also part of the wMTrck plugin and has been used for different organisms including *C.elegans* and *Drosophila* [54,55]. We found that the larval shape is highly dependent on the crawling behaviour, especially crawling direction. Larvae possess an elongated shape upon straight forward movement, and an

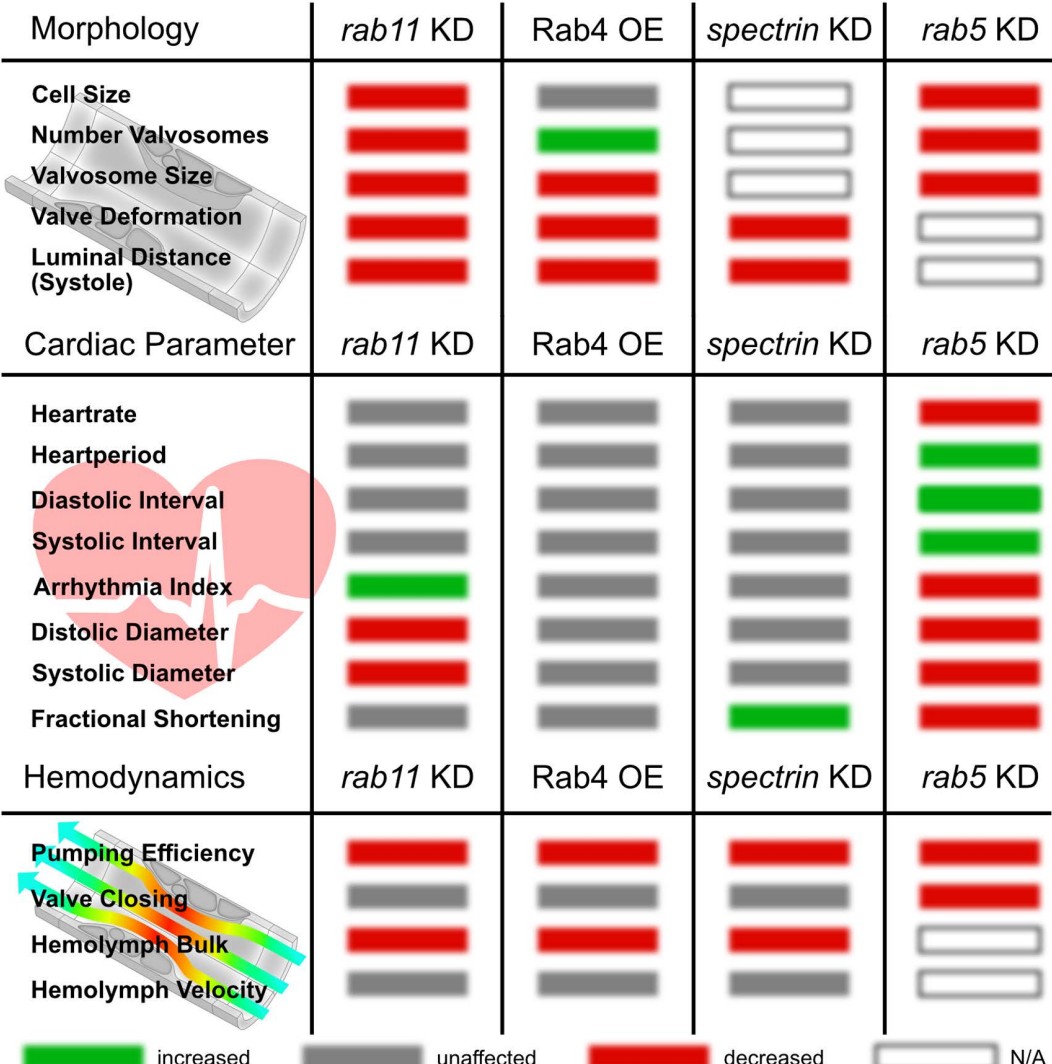

| Morphology | *rab11* KD | Rab4 OE | *spectrin* KD | *rab5* KD |
|---|---|---|---|---|
| Cell Size | decreased | unaffected | N/A | decreased |
| Number Valvosomes | decreased | increased | N/A | decreased |
| Valvosome Size | decreased | decreased | N/A | decreased |
| Valve Deformation | decreased | decreased | decreased | N/A |
| Luminal Distance (Systole) | decreased | decreased | decreased | N/A |

| Cardiac Parameter | *rab11* KD | Rab4 OE | *spectrin* KD | *rab5* KD |
|---|---|---|---|---|
| Heartrate | unaffected | unaffected | unaffected | decreased |
| Heartperiod | unaffected | unaffected | unaffected | increased |
| Diastolic Interval | unaffected | unaffected | unaffected | increased |
| Systolic Interval | unaffected | unaffected | unaffected | increased |
| Arrhythmia Index | increased | unaffected | unaffected | decreased |
| Distolic Diameter | decreased | unaffected | unaffected | decreased |
| Systolic Diameter | decreased | unaffected | unaffected | decreased |
| Fractional Shortening | unaffected | unaffected | increased | decreased |

| Hemodynamics | *rab11* KD | Rab4 OE | *spectrin* KD | *rab5* KD |
|---|---|---|---|---|
| Pumping Efficiency | decreased | decreased | decreased | decreased |
| Valve Closing | unaffected | unaffected | unaffected | decreased |
| Hemolymph Bulk | decreased | decreased | decreased | N/A |
| Hemolymph Velocity | unaffected | unaffected | unaffected | N/A |

increased | unaffected | decreased | N/A

**Fig 7. Cardiac valve malformation impacts morphology, cardiac parameter and hemodynamics.** Upon expression of Rab4 or knockdown of *rab11*, *rab5* and *spectrin* valve cell's morphology is severely affected [27]. Knockdown of *rab11* results in reduced diastolic and systolic diameters and an increased arrhythmia index. Ectopic expression of Rab4 increases the number of valvosomes and reduces their size, but cardiac parameter were unaffected. Knockdown of *spectrin* reduced valve cells deformation capability and luminal distance, as well as fractional shortening. Downregulation of *rab5* negatively affected cell and valvosomal size, as well as number of valvosomes. In addition, heartrate, diastolic and systolic diameter and fractional shortening were significantly reduced (see also S1 Fig). Cardiac malformation leads to reduced cardiac pumping efficiency and hemolymph distribution. In addition, significantly less hemolymph is ejected per heartbeat, while hemolymph velocity and valve luminal closure are unaffected.

increasingly roundish shape as they bend or turn (Fig 8F). Analysis of larval body shapes point to a more roundish shape in the Rab4 overexpressing and *alpha spectrin* knockdown lines (Fig 8G). In contrast, downregulation of *rab5* or *rab11* had no significant impact on larval shape. Both groups displayed a predominantly elongated shape, similar to controls (Fig 8G).

In sum, our results point to a reduced physical activity in valve mutants, displayed by a reduced crawled distance and reduced crawling speeds. In addition, larvae tend to bend their body and change direction more often, when valves are malformed.

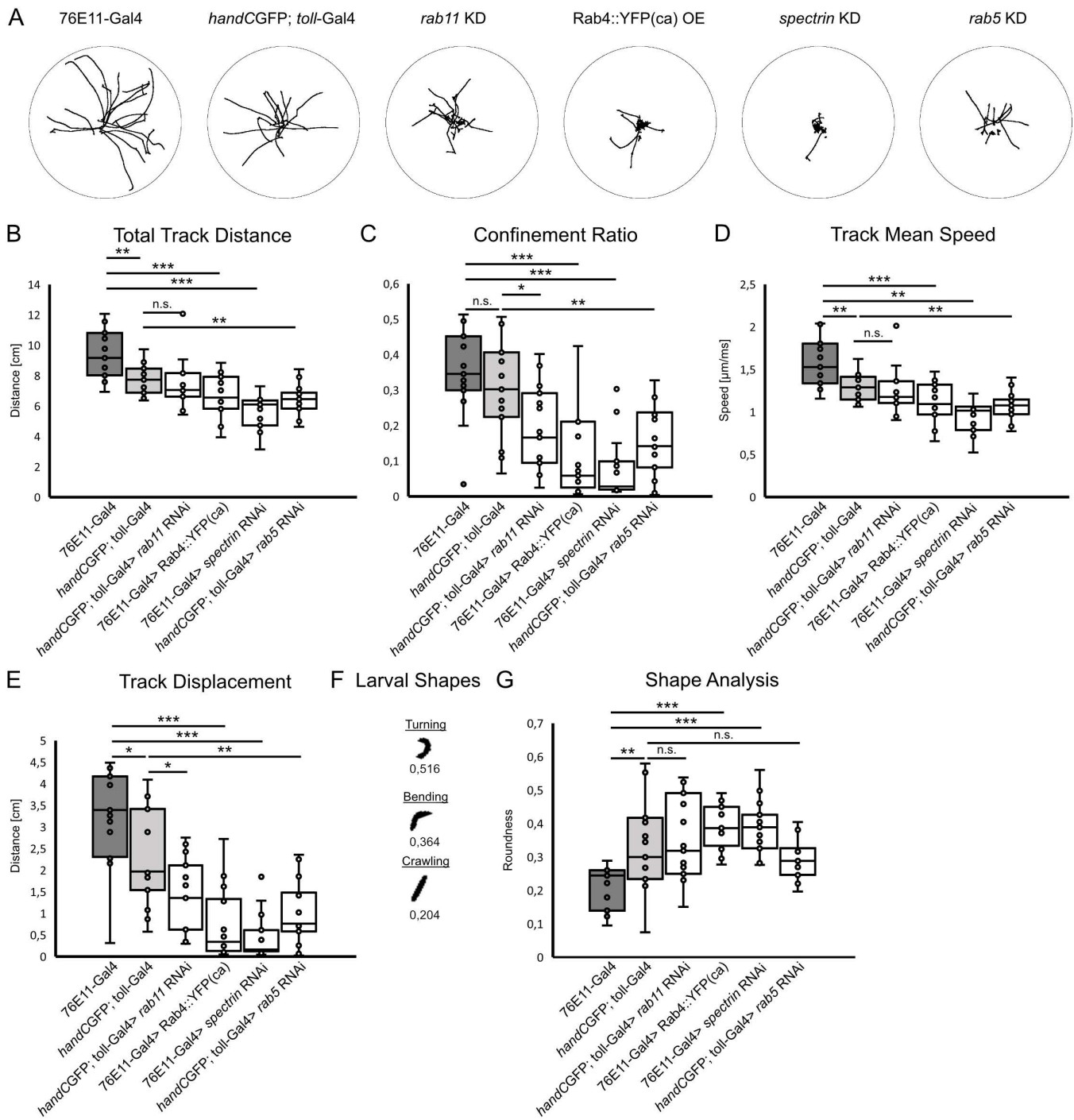

**Fig 8. Cardiac valve malformation impacts crawling and movement of larvae.** (A) Schemes illustrating crawling tracks of larvae upon *spectrin*, *rab11* and *rab5* knockdown, and ectopic expression of Rab4. (B) The total distance crawled is significantly reduced in *spectrin* and *rab5* knockdown animals and in Rab4 overexpression lines (Two-tailed Student's *t*-test), while *rab11* knockdown had no effect (*Mann-Whitney* test). (C) Confinement ratio and mean track speed (D) are significantly reduced in all animals with damaged valves investigated lines (Two-tailed Student's *t*-test), except for *rab11* knockdown lines, where track mean speed is unaffected (D, *Mann-Whitney* test). (E) Track displacement is significantly reduced, if valves are malformed. (F) Scheme showing different types of larval's body shape upon crawling and respective roundness values (0 = elongated, 1 = round). (G) Knockdown of *rab11* and *rab5* had no impact on larval's shape but knockdown of *spectrin* and overexpression of Rab4 led to an increased roundish shape of larvae (Two-tailed Student's *t*-test). *N* = 15 animals per genotype *$P < 0.05$, **$P < 0.01$, ***$P < 0.001$.

## Discussion

Circulatory systems, with muscular pumps as the driving force for blood or hemolymph transport, constitute at the center of major physiological processes in animal and human bodies. While the histology and morphology of vertebrate hearts and related structures have been studied for centuries, the circulatory systems of invertebrates, and especially of arthropods with their unique open circulatory system, has become increasingly attractive for scientists only in the last few decades. The reason for this is undoubtedly the realisation that numerous fundamental biological processes can be analysed very efficiently in model organisms that offer a simpler morphology combined with superior genetic amenability. The most important functions of the insect circulatory system include delivering nutrients and hormones, removing waste products, coordinating defense mechanisms, modulating heat transfer, assisting in gas exchange and many more processes that depend on proper fluid distribution. To achieve this task, a variety of different systems, muscular pumps and associated structures have evolved. It is assumed that a first 'heart' was already present in an early bilaterian ancestor. Constituting simple tubular and pulsatile structures, these heart primordia lacked cardiac chambers, septae, valves or enclosed vascular systems. Thus, fluid flow was likely unidirectional and presumably inefficient [56]. Today, the only insects lacking a pulsatile cardiac structure are microinsects, like the members of the Ptiliidae, Trichogrammatidae and Mymaridae [57]. However, it is unclear whether they lost their cardiac systems as a result of allometry, the downscaling of organs as the body size shrinks. Due to a highly reduced body size, solute transport by diffusion is likely sufficient.

Throughout evolution, intracardiac valves as integral constituents of the insect heart tube appeared, and were thought to regulate and direct fluid transport in circulatory systems and thus boost efficiency and performance. Among insect families, valves are found in different numbers, sizes and functionalities but, in all cases, constitute some kind of gating structures. Outside circulatory systems, valves are also found in renal tubes, lymphatic vessels or the digestive tract [8,16–18,58,59]. Interestingly, in humans cardiac valve damage has severe consequences and leads to, among others, stenosis affecting heart rate [45], cardiac output [15], altered heart morphology and negatively impacts the whole vascular system [12,42–44,60]. The main characteristic of cardiac stenosis is a reduced luminal opening due to a reduced deformation capability of cardiac valves [37]. Interestingly, we now found similar characteristics and phenotypes in the *Drosophila* heart (this study; [27]).

We showed that a reduction in deformation capability of valve cells leads to a reduced luminal opening, resulting in decreased transport and distribution of hemolymph in the body of the animal and a consequently impaired heart performance. The ability to permit an unobstructed flow depends on the mobility and flexibility of cardiac valves, which is severely impaired upon valve malformation (this study; [7]). Of note, blood flow analysis and especially detailed analysis of aortic blood velocity are of particular interest in human clinical cardiac research and one of the key indicators of a developing aortic stenosis [61,62]. Herein, we show that functional valves enable proper luminal opening and unobstructed fluid flow upon heart beating. In contrast, the fly heart pumps against a narrow luminal opening, caused by damaged valves, which cause obstructed flow and increased acceleration of hemolymph at the heart lumen close to the valve. In fact, in humans it was shown that the pressure gradient across the aortic valve increases exponentially with a decreasing luminal valve opening, which leads to changes in hemodynamics and higher streaming velocities due to a narrow opening [15,37,63]. Interestingly, the dorsal vessel of insects is regarded as a relatively weak organ, which cannot pump against a larger amount of pressure compared to the human heart [64–66]. Thus, upon valve damage, the heart cannot pump a sufficient amount of hemolymph through less deforming valve cells, resulting in a significantly reduced hemolymph distribution in *Drosophila* (Fig 2).

A detrimental effect of a reduced hemolymph distribution on animal fitness appears likely. Physical activity is highly beneficial for cardiovascular health and may prevent age-related cardiac dysfunction in flies and men [67–70]. In contrast, cardiac diseases often lead to reduced cardiac performance and higher exercise intolerance [60]. Interestingly, hemolymph distribution in *Drosophila* is primarily driven by the activity of the heart, but also by the overall contraction activity of the body wall muscles [52,53]. Animals lacking a fully functional heart indeed show reduced fitness or limited lifespan, e.g., seen in *pericardin* or *lonely heart* mutants [31,71,72]. Moreover, many genes affecting heart differentiation led to similar phenotypes if the flies are treated by stress, e.g., a higher temperature [73]. Our present study focused on the contribution of the valve cells to heart performance and fitness. We found that a reduced heart performance caused by damaged valves negatively impacts endurance of larvae. In addition, the animals may try to compensate loss of circulation efficiency by increasing body movement, i.e., bending and turning.

We conclude that upon evolution intracardiac valves evolved to boost cardiac performance, i.e., to enhance hemolymph distribution in insect hearts. In *Drosophila* intracardiac valves developed by modulation of already existing molecular pathways, generating a highly specialised cell with unique cellular organelles. We showed that this structure is ideally suited to fulfil its main tasks:

1) Effectively seal the heart lumen upon diastole, enabling proper sucking in of hemolymph into the heart chamber.

2) Effectively deform upon systole, enabling unobstructed hemolymph flow and distribution by fully opening the heart lumen.

This idea is further supported by studies pointing to cardiac valves as hemodynamic valves [30]. The resistance of the valve decreases at a certain pressure and finally leads to the opening of the heart lumen. The presence of the valvosomal compartment enables valve cells to deform properly upon hemodynamic pressure. In contrast, the absence of valvosomes will increase valve pressure resistance and reduce their deformation capability, resulting in a narrow lumen. As a result, the heart cannot pump efficiently against this higher pressure and reduced luminal opening.

The term "valve" stands for a dedicated structure that enables unidirectional fluid flow by a mechanism that is reminiscent flaps. Flaps close a lumen as the fluid flows in the opposite direction against the flap. This is not the case for the intracardiac valve cells in *Drosophila*. We believe that the primary function of these valves is not to represent a "one-way-valve", but indeed a hemodynamic valve as previously suggested [30]. This hypothesis is now supported by various observations presented in this and previous studies:

1) The absence of dye in the aorta at the end of systole, indicates no hemolymph pressure anterior to and against the valves. The absence of a backward orientated pressure against the valve [30].

2) When larvae are dissected to observe the heartbeat, the body interior and the interplay of muscles, body wall movement, cardiac activity and circulation are lost. Under these circumstances we observed a periodically occurring retrograde flow. However, this retrograde flow does not lead to a luminal heart closure by the valves [26].

3) The larval cardiac valve persists into adulthood. Two additional valves, each consisting of two individual valve cells, are formed during metamorphosis and all valves have a similar morphology as the larval intracardiac valve [25,28]. However, in adult *Drosophila,* valves permit a periodically occurring retrograde flow [22,74,75].

4) Bidirectional hemolymph flow is reported for a variety of insect species and is based on additional contractile regions within the heart in combination with anteriorly and posteriorly located excurrent openings [22,74–76]. It has been speculated that bidirectional hemolymph flow might be beneficial for cleaning the luminal compartment of the heart from cell debris and other material remains [77].

Although a number of questions remain to be addressed, the present study highlights for the first time the necessity and physiological importance of cardiac valves in an invertebrate model organism. Hereby, we present a valuable set of tools and methods for future studies on hemodynamics in insects and on the role of valves cells in circulation.

## Materials and methods

### Fly stocks and genetics

The *toll*-Gal4 and *hand*C-GFP lines used in this study were made by our laboratory [78].

The following lines were obtained from the Bloomington *Drosophila* Stock Center (BDSC) at Indiana University: 76E11-Gal4 (RRID:BDSC_39933), UAS-Rab4pQ67L (RRID:BDSC_9770) and UAScd4td::GFP (RRID:BDSC_35836). The following RNAi lines were obtained from the VIENNA *Drosophila* Resource Center (v) or Bloomington Drosophila Stock Center (BL): rab5 v103945, rab11 v108382 and spectrin v42053. As a control strains respective valve specific drivers toll-Gal4 and 76E11-Gal4 were used. The expression patterns of the Gal4-driver lines used herein were previously described: *toll* enhancer [26] and 76E11 [79] and [80]). Fly husbandry was carried out as described previously [81].

### Antibodies and reagents

Antibodies were used to detect Spectrin (1:20; Developmental Studies Hybridoma Bank, USA), Monoclonal rabbit anti-GFP (1:2000) was from Abcam (Ab6556) and monoclonal mouse anti-GFP (3E6, 1:500) was from Invitrogen (A-11120; Thermo Fisher Scientific). Secondary antibodies used were anti-mouse Cy2, anti-mouse Cy3 and anti-rabbit Cy2 (1:200; Dianova, Germany).

### Immunostaining of valve cells of third instar larvae

Third instar wandering larvae were dissected following previously published protocols [25,27,82]. Briefly, animals were pinned down on their dorsal side on a Sylgard plate, covered with PBS, opened ventrally and Viscera were carefully removed. Specimens were fixed in 4% methanol-free paraformaldehyde. After washing steps, samples were permeabilised with 1% Triton in PBS, followed by three additional washing steps. Afterwards, nonspecific epitopes were blocked by incubation in saturation buffer. Primary antibodies were diluted in PBS buffer and incubated overnight at 4 °C under constant shaking before the solution was removed and replaced by BBT buffer for thorough washing. Secondary antibodies, with coupled fluorophores, were diluted in PBS and incubated at room temperature in the dark for 2 h. Unbound antibodies were removed by three washing steps in PBS. Finally, samples were embedded in Fluoromount-GTM (Thermo Fisher Scientific) with DAPI or RotiMount FluorCare with DAPI (Roth) and imaged with a Zeiss LSM 800 laser scanning microscope. Images were analysed using Fiji ImageJ software.

### *Dye angiography* and hemolymph accumulation

The basic protocol was modified after [30] and [31]. Briefly, selected white pre-pupae were glued on a microscope slide using double sided Scotch tape. Injections were performed using small custom made glass capillaries equipped on a micro-manipulator connected to an

Eppendorf FemtoJet micro-injection system. Injection of black ink (Pelican) into the posterior body part was done by a single injection for 0,1 sec at 2 psi. Accumulation of the dye was recorded using a Leica WILD MZ8 stereomicroscope equipped with a Basler acA2000-165uc camera and transmitted light illumination. Videos were captures for 30 seconds at a frame rate of ~100 frames/s and converted to 8-bit grayscale videos in ImageJ. Background was substracted by deducting pixel intensities from the first video frame from all following frames. Accumulation of dye was analyzed in an anterior 15x15 pixel ROI using ImageJ and the "Plot Z-axis profile"-tool. 10 animals per genotype were analyzed and the mean dye accumulation was estimated. Data were analyzed using an unpaired two-tailed Student's *t*-test.

### Valve closing capability

Valve closing capability was determined from dye angiography videos (see section above). In brief a 15 x 15 pixel ROI was set anterior to the valve cells and pixel intensity was measured for five seconds with ImageJ using the *Plot z-axis*-tool. Background intensity is mostly influenced by the individual darkening of the animal's cuticle and was thus calculated for each animal prior to dye injection, respectively. Each time the dye bulk passes the ROI and streams anteriorly, this leads to a time-dependent peak increase in pixel intensity, which afterwards goes down to a baseline, if valves close properly.

### Hemolymph bulk analysis

For hemolymph bulk analysis videos from dye angiography were taken and processed with ImageJ. Mean hemolymph bulk pixel Intensity was measured from single dye packages within the aorta, using ImageJ *Wand-Tracing*-Tool to outline the entire area of the dye package. Five animals per genotype were evaluated and for each animal five dye bulk packages were analyzed, respectively. Data were analyzed using an unpaired two-tailed Student's *t*-test.

### Hemolymph aorta velocity

Hemolymph velocity in 3rd larval aorta was determined from dye angiography analysis and as previously described [30]. In brief, two ROI with a size of 15 x15 Pixel were set in the posterior and anterior part of the aorta, respectively. Velocity of hemolymph was calculated by measuring the distance between the ROI divided by the time it takes for one hemolymph bulk to reach peak pixel intensity in both ROI. For each genotype 10 larvae and 3 cardiac cycles were analyzed and mean velocities per animal were calculated. Data were analyzed using an unpaired two-tailed Student's *t*-test and *Mann-Whitney* test.

### Valve luminal distances and opening time

For analysis highspeed videos of semi-intact heart preparations were used. Luminal distances of valves were calculated manually by measuring the minimum luminal distance between opposing valve cells in the fully open state in a 90° angle and from the cytoplasmic border of valve cells. For each genotype 8 animals and 5 heartbeats were analysed, for *rab5* knockdown lines 5 animals were analysed. Valve opening time was measured by calculating the time between two closed valve states, respectively. For each animal mean luminal opening time was calculated from 10 single heartbeats and 10 animals per genotype were analyzed. Data were analyzed using an unpaired two-tailed Student's *t*-test.

### Hemolymph velocity at valves

For analysis of streaming velocities at valve level, we injected red fluorescent polystyrene particles with a diameter of 1,2 μm (2% w/v in artificial hemolymph; microParticles GmbH, Berlin,

Germany) with the same setup used for dye injection experiments described above. Particle flow was recorded using a Zeiss LSM 5 Pascal confocal microscope equipped with a Basler acA800-510uc camera and an ebq 100 light lamp with adequate filter sets. Highspeed videos at 200 fps were captured for 10 seconds. Particles were manually tracked and velocities analyzed with ImageJ and the *Manual Tracking with Trackmate*-Plugin. Position of valves upon heart beating was estimated from *hand*C-GFP reporter signal and particle streaming velocities of all tracks within the valve region were estimated. Data were analyzed using nonparametric *Mann-Whitney-* test.

## Cardiac myofiber organization

Wandering 3[rd] instar larvae were collected and prepared in PBS containing 8% $MgCl_2$, fixed for 1 h in 4% Formaldehyde at room temperature, permeabilized for 1 h in 1% Triton PBS prior to incubation for 1 h at RT in 1:100 Phalloidin TRITC solution. Images were captured with a Zeis LSM800 confocal microscope. Myofiber density and orientation were analyzed in a single 100 μm x 50 μm ROI in abdominal segment five (A5) between pairs of functional ostia cells. Myofiber density was estimated by threshold-based measurements of the heart area in relation to myofiber coverage, respectively. Myofiber orientation was measured in ImageJ using *Directionality*-tool and Fourier-components method with default settings. Heart Area was calculated from orthogonal sections acquired from confocal stacks using ImageJ *orthogonal view* -function. Sections from the abdominal segment A5 were taken and analyzed with ImageJ *polygon* selection-tool to outline heart area. N = 10 animals per genotype were investigated and data were analyzed using an unpaired two-tailed Student's *t*-test.

## *Analysis of cardiac parameter using SOHA-method*

Semi-intact heart preparations were done as previously described [26,46]. In brief, third instar wandering-larvae were dissected in artificial hemolymph from the ventral side and viscera was removed [83]. Specimens were allowed to recover for at least 5 min. Videos were captured with Basler piA640-210gm high-speed camera at 200 frames $s_{-1}$, mounted on an Olympus BX41TF stereomicroscope. Recordings were done using *pylon-Viewer* software (Basler AG) at 23 °C to ensure constant conditions for all specimens. Videos were further processed with ImageJ and heart parameters were examined via semi-automatic optical heartbeat analysis (SOHA; [46,84]); N = 10 animals per genotype were investigated, for *rab5* knockdown lines five animals, and data were analyzed using an unpaired two-tailed Student's *t*-test.

## Larval crawling assay

Larval crawling analysis was done as previously described [51]. In brief, wandering third instar larvae were collected from tubes and placed individually on a 92 x 16 mm polystyrene, humidified petri dish. Positioning of larvae causes short paralysis and immobilization of larvae, therefore imaging was started after the first initial head movements were noticed. Crawling larvae were imaged for 60 seconds using a Nikon D5200 digital camera with a 40 mm 1:2,8 mm Nikon AF-S Micro NIKKOR objective equipped to Kaiser RS1 copystand with scale and provided with four 3000 K opal lamps. Larvae that reached the end of the petri dish and started to climb up the wall during recording, were discarded. Videos were further processed and analyzed using ImageJ and the *Trackmate*-plugin [85]. For larval shape analysis ImageJ's *Shape Descriptors* and threshold-based measurements were performed. The major and minor axis of a fitted ellipse describing the larvals shape was calculated for each frame. Roundess

of larval shape was then calculated as the inverse of the aspect ratio: (4*area/($\pi$*major axis$^2$). Values approaching 0,0 indicate an increasingly elongated shape [54,55]. For each genotype 15 larvae were analysed and data were analyzed using an unpaired two-tailed Student's *t*-test.

## Supporting information

**S1 Fig.  Cardiac valve malformation impacts cardiac parameter.** Knockdown of *rab11* results in reduced diastolic and systolic diameters and an increased arrhythmia index. Ectopic expression of Rab4 increases the number of valvosomes and reduces their size, but cardiac parameter were unaffected. Knockdown of *spectrin* reduced valve cells deformation capability and luminal distance, as well as fractional shortening. Downregulation of *rab5* negatively affected cell and valvosomal size, as well as number of valvosomes. In addition, heartrate, diastolic and systolic diameter and fractional shortening were significantly reduced. Cardiac malformation leads to reduced cardiac pumping efficiency and hemolymph distribution. N = 10 animals per genotype were investigated, for *rab5* knockdown lines five animals, and data were analyzed using an unpaired two-tailed Student's t-test *P < 0.05, **P < 0.01, ***P < 0.001. (TIFF)

**S1 Data.  In the Excel table provided, all primary data measured and analyzed for the results presented here are listed according to the corresponding figure table.** (XLSX)

**S1 Video.  3rd Instar Drosophila larval heart rhythmically beating.** Inflow tracts (ostia cells) upon systole due to cardiac chamber contraction. Intracardiac valves open during systole due to an increased forward-orientated hemolymph pressure. (MP4)

**S2 Video.  Scheme illustrating the methodology of dye angiography analysis.** Black ink is injected in the posterior end of early white prepupae. Tracer dye accumulates over time in the anterior region due to the continuous pumping activity of the heart. (MP4)

## Acknowledgements

We thank Martina Biedermann, Kerstin Etzold and Mechthild Krabusch for excellent technical assistance and the Bloomington *Drosophila* Stock Center and Vienna *Drosophila* Resource Center for providing stocks essential for this work. Furthermore, we thank our student Maike Spielmeyer for supporting us with experimental preparation and imaging. We also thank PD Dr. Heiko Harten and Dr. Maik Drechsler for various experimental support and critical reading of the manuscript.

## Author contributions

**Conceptualization:** Christian Meyer, Achim Paululat.

**Formal analysis:** Christian Meyer, Achim Paululat.

**Funding acquisition:** Achim Paululat.

**Investigation:** Christian Meyer.

**Methodology:** Christian Meyer.

**Project administration:** Achim Paululat.

**Supervision:** Achim Paululat.

**Validation:** Christian Meyer.

**Visualization:** Christian Meyer.

**Writing – original draft:** Christian Meyer, Achim Paululat.

**Writing – review & editing:** Christian Meyer, Achim Paululat.

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
