## [Decision Letter · Decision Letter 0]

20 Jan 2025

PGENETICS-D-24-01467

Valve cells are crucial for efficient cardiac performance in Drosophila

PLOS Genetics

Dear Dr. Paululat,

Thank you for submitting your manuscript to PLOS Genetics. You will see that the reviewers were overall very positive about your work. Nevertheless, Reviewer 1 raised some important concerns that you would need to address before it can be considered for publication in PLOS Genetics. We therefore invite you to submit a revised version of the manuscript that addresses the points raised by all reviewers.

Please submit your revised manuscript within 60 days Mar 21 2025 11:59PM. If you will need more time than this to complete your revisions, please reply to this message or contact the journal office at plosgenetics@plos.org. Please include the following items when submitting your revised manuscript:

We look forward to receiving your revised manuscript.

Kind regards,

John Ewer

Academic Editor

PLOS Genetics

Pablo Wappner

Section Editor

PLOS Genetics

Aimée Dudley

Editor-in-Chief

PLOS Genetics

Anne Goriely

Editor-in-Chief

PLOS Genetics

**Journal Requirements:**

https://journals.plos.org/plosgenetics/s/submission-guidelines#loc-parts-of-a-submission

- TM on page: 27.

5) We notice that your supplementary Figures are included in the manuscript file. Please remove them and upload them with the file type 'Supporting Information'. Please ensure that each Supporting Information file has a legend listed in the manuscript after the references list.

Potential Copyright Issues:

i) Please confirm that you are the photographer of Figure 3B, or provide written permission from the photographer to publish the photo(s) under our CC BY 4.0 license.

ii) Figures 1, 2, 3, and 7; Please confirm whether you drew the images / clip-art within the figure panels by hand. If you did not draw the images, please provide a link to the source of the images or icons and their license / terms of use; or written permission from the copyright holder to publish the images or icons under our CC BY 4.0 license. Alternatively, you may replace the images with open source alternatives. See these open source resources you may use to replace images / clip-art:

7) Please amend your detailed Financial Disclosure statement. This is published with the article. It must therefore be completed in full sentences and contain the exact wording you wish to be published.

**Reviewers' comments:**

Reviewer's Responses to Questions

**Comments to the Authors:**

Reviewer #1: Review:

Valve Cells are Crucial for Efficient Cardiac Performance in Drosophila., by Chris Meyer and Achim Paululat.

Summary: This is a very interesting set of studies describing an under-studied component of Drosophila heart function, the intracardiac valves. The authors provide several new methods to describe and quantify valve function and show that valve morphology and function impact overall cardiac physiology. Figure 7 is an especially well-designed and user-friendly summary of the findings. This study should be of broader interest to researchers studying the connections between hemodynamic flow and cardiac performance.

While there are minor language and technical issues that need to be corrected there are also some significant issues with the presented data that will need to be addressed before the results can be fully vetted.

Major:

The study provides a comprehensive overview of hemolymph flow and cardiac valve function in Drosophila but would benefit from a discussion of why the authors chose to study these particular genes of interest.

Overall, additional detail is required in order to fully understand the methodologies used and assess the results presented.

1.) Fig. 1 - This figure legend indicates that manipulating Rab5, which facilitates migration of valve cells to their final location in the lumen via endocytosis, and Rab4/11 which regulate the recycling of endosomes responsible for valve cell transport, leads to smaller valve cells and absent valvosomes. These differences should be quantified and included in this figure.

Although the schematics shown in D’-H’ could be useful they don’t correspond to what is shown in the actual micrographs D-H.

The authors should elaborate on the methodology used in Figure 1. While the phenotype is clearly depicted, the text currently offers only a superficial reference to a previous paper. More detailed methods, quantifications and explanations should be included for the reader of the current article.

2.) Fig. 2,3,4,6,8, S1 – How the authors used the two different drivers (76E11-Gal4 and Toll-gal4) is unclear. The authors should clearly indicate which driver was used to modulate expression when crossed to the different rab and spectin lines. Ideally, each driver should be crossed to ALL of the KD or OE lines and the results shown for each driver set. This would ensure that results are consistent regardless of driver, i.e. in differing genetic backgrounds. Were additional KD lines tested for each gene?

3.) The expression patterns for each driver line should be documented and/or cited. In particular it is not clear where 76E11-Gal4 is expressed, the available literature indicates it is a neuronal driver, how does that fit with this experiment on valves? Is the Toll-Gal4 expression exclusively in valves and no other tissues? Was the extent of KD quantified?

4.) For data in Fig. 2 & 3, how was the pixel intensity data normalized to allow comparisons between genotypes? For example, peak intensities for controls in Fig. 3 D & E are ~7 to 8, however, peak intensities for Rab11 KD and spectrin KD (F & H) are higher at ~ 10 to 11; yet the quantification shown in J indicates significant reductions in flies with Rab11 or spectrin KD compared to controls.

5.) The conclusions in Figure 6 are inconsistent with Supplemental Figure 1, where diastolic diameter is reduced in Rab11 and Rab5 KD models.

5.) More detail should be provided as to how the densities and myofibril orientation were obtained in Fig. 6. Did the authors analyze specific ROIs within the heart, if so where, how many / heart? Show examples of the myofibrillar organization in the KD and OE lines.

5.) For data presented in all Graphs (except Fig. 5G, where only two groups are being compared) the appropriate statistical test would be a one-way ANOVA.

6.) Given the large SEMs in most of the graphs there appears to be a very large amount of variability in these samples (including even in the controls), making some of the indicated significant differences rather questionable and indicated significant differences will likely change when the correct statistical tests are applied. Graphs should show all data points so the reader can get a feel for this variability. Because physiological data can be variable this submission would benefit from higher Ns for most of the analyses.

Minor:

1. Fly stock codes do not need to be included in the figures themselves. These should be specified in a separate table in Methods or supplemental data. This will make interpretation of the genotypes in the graphs a little easier.

2. The specific driver line crossed to each rab and spectrin line should be indicated. It is not clear from the graphs that Rab4 is an overexpression line.

3. Figures 2 and 3 could potentially be reduced and combined to minimize redundancy (Fig. 3a for example does not provide more info than is available in Fig. 3b) and improve flow.

4. Fig. 4, It is unclear how the luminal distance was determined for valves that are not completely juxtaposed. It would be helpful to include a line showing where this distance is measured. In addition to control images shown in G-K, authors should include examples of at least one of the KD or OE lines.

5. Fig. 5, depicts an interesting analysis of blood flow velocities but why is only one gene of interest included in this analysis since all the genes tested seem to affect valves and flow in some way? It is unclear what is being shown in the red box in (B) and what is narrowing.

6. Fig. 6, the data for Rab5 appears inconsistent with an n=10 sample size based on the error bars. This discrepancy warrants further clarification.

7. Fig. 6, please provide more explanation of what is shown in A’,B’ and C’, please show examples of the myofibrillar structure from the spectrin and Rab5 lines.

Reviewer #2: In the present manuscript Christian Meyer and Achim Paululat analyse in an unprecedented way and using a large set of innovative and elegant approaches the role of cardiac valves. They take advantage of in house generated unique tools that allow them to manipulate genetically valve properties and to determine their role in wondering larva in heart function, structure, in circulatory system performance and in animal fitness as a whole. Experiments are performed with rigor and presented data are of quality. Angiography and hemolymph velocity approaches are of high interest to the field as could be applied for defining cardiac performance in different genetic contexts and in cardiac diseases modelling. Manuscript is well written and conclusions fully supported by the presented data, with illustration of main findings in Fig. 7.

I have only minor comments presented below.

Minor comments:

1. In Figure 1 authors present a summary of valve phenotypes observed in a serie of identified previously mutants and different valve targeting GAL4 lines. It would be important to provide more information about the valve driver lines and specificity of their expression and explain the reason two different drivers are used to generate valve phenotypes in Fig. 1 but also in following experiments.

2. Both Rab11 and Rab 5 KD lead to a loss of valvosomes, however Rab5KD impact's on heart parameters is much more severe than that of Rab11KD. This needs to be discussed especially with respect to hemolymph velocity (Fig. 3K).

Reviewer #3: Review

The manuscript “Valve cells are crucial for efficient cardiac performance in Drosophila” by Christian Meyer and Achim Paululat investigates the physiological consequences of cardiac valve malformations in Drosophila. They focus on the intracardiac valve between the aorta and ventricle, which regulate hemolymph flow in the larval fly heart. The find that valve malformations reduce hemolymph transport efficiency as measured by dye angiography, resulting in decreased cardiac output. Using genetic manipulations targeting Rab5, Rab11, Rab4, and alpha-Spectrin in valve cells they provide a model for studying valve-related pathologies and cardiac performance in a simple system.

This study addresses a significant gap in understanding Drosophila cardiac physiology by functionally analyzing the role of the valve cells in heart physiology, and which highlights an evolutionary innovation in the intracardiac valves of Drosophila, providing a basis for comparative studies.

Overall, I do not see any overall weaknesses in the manuscript. One might broaden the discussion to include implications for vertebrate cardiology and potential conservation of underlying mechanisms, and providing more direct evidence for such a similarity would certainly increase the impact.

Weaknesses:

Statistics: change all bar graphs to box/whisker and show all data points; t-tests require normally distributed data - was this ensured? For non-normal data Wilcoxon tests are warranted.

Crawling Assay Interpretations: The link between reduced crawling and valve malfunction could be influenced by external factors not accounted for, such as genetic pleiotropy. Both controls show already differences between each other. I am not sure about the significance of this experiment.

**Have all data underlying the figures and results presented in the manuscript been provided?**

Reviewer #1: **No: **

Reviewer #2: Yes

Reviewer #3: Yes

PLOS authors have the option to publish the peer review history of their article (what does this mean? ). If published, this will include your full peer review and any attached files.

**Do you want your identity to be public for this peer review?** For information about this choice, including consent withdrawal, please see our Privacy Policy .

Reviewer #1: No

Reviewer #2: No

Reviewer #3: No

**Figure resubmission:**
---

## [Decision Letter · Decision Letter 1]

7 Feb 2025

Dear Dr Paululat,

We are pleased to inform you that your manuscript entitled "Valve cells are crucial for efficient cardiac performance in Drosophila" has been editorially accepted for publication in PLOS Genetics. Congratulations!

Yours sincerely,

John Ewer

Academic Editor

PLOS Genetics

Pablo Wappner

Section Editor

PLOS Genetics

Aimée Dudley

Editor-in-Chief

PLOS Genetics

Anne Goriely

Editor-in-Chief

PLOS Genetics

Comments from the reviewers (if applicable):

Reviewer's Responses to Questions

**Comments to the Authors:**

Reviewer #2: Authors provided compeling and satisfactory answers to the raised points. I have no more comments.

Reviewer #3: I enjoyed reading this updated manuscript an I am looking forward to it publication. No further comments.

**Have all data underlying the figures and results presented in the manuscript been provided?**

Reviewer #2: Yes

Reviewer #3: Yes

PLOS authors have the option to publish the peer review history of their article (what does this mean? ). If published, this will include your full peer review and any attached files.

**Do you want your identity to be public for this peer review?** For information about this choice, including consent withdrawal, please see our Privacy Policy .

Reviewer #2: No

Reviewer #3: No

**Data Deposition**

http://datadryad.org/submit?journalID=pgenetics&manu=PGENETICS-D-24-01467R1

**Press Queries**

---

## [Editor Report · Acceptance letter]

PGENETICS-D-24-01467R1

Valve cells are crucial for efficient cardiac performance in Drosophila

Dear Dr Paululat,

We are pleased to inform you that your manuscript entitled "Valve cells are crucial for efficient cardiac performance in Drosophila" has been formally accepted for publication in PLOS Genetics! Your manuscript is now with our production department and you will be notified of the publication date in due course.

With kind regards,

Anita Estes

PLOS Genetics

On behalf of:
